# Cross-Domain Transfer via Semantic Skill Imitation

**Karl Pertsch**[1][*], **Ruta Desai**[2], **Vikash Kumar**[2],
**Franziska Meier**[2], **Joseph J. Lim**[3], **Dhruv Batra**[2,4], **Akshara Rai**[2]

[1]University of Southern California, [2]Meta AI, [3]KAIST, [4]Georgia Tech
https://kpertsch.github.io/star

**Abstract:** We propose an approach for *semantic* imitation, which uses demonstrations from a source domain, e.g., human videos, to accelerate reinforcement learning (RL) in a different target domain, e.g., a robotic manipulator in a simulated kitchen. Instead of imitating low-level actions like joint velocities, our approach imitates the sequence of demonstrated semantic skills like "opening the microwave" or "turning on the stove". This allows us to transfer demonstrations across environments (e.g., real-world to simulated kitchen) and agent embodiments (e.g., bimanual human demonstration to robotic arm). We evaluate on three challenging cross-domain learning problems and match the performance of demonstration-accelerated RL approaches that require in-domain demonstrations. In a simulated kitchen environment, our approach learns long-horizon robot manipulation tasks, using less than 3 minutes of human video demonstrations from a real-world kitchen. This enables scaling robot learning via the reuse of demonstrations, e.g., collected as human videos, for learning in any number of target domains.

**Keywords:** Reinforcement Learning, Imitation, Transfer Learning

## 1 Introduction

Consider a person imitating an expert in two scenarios: a beginner learning to play tennis, and a chef following a recipe for a new dish. In the former case, when mastering the basic skills of tennis, humans tend to imitate the precise arm movements demonstrated by the expert. In contrast, when operating in a familiar domain, such as a chef learning to cook a new dish, imitation happens on a higher scale. Instead of imitating individual movements, they follow high-level, semantically meaningful skills like "stir the mixture" or "turn on the oven". Such *semantic skills* generalize across environment layouts, and allow humans to follow demonstrations across substantially different environments.

Most works that leverage demonstrations in robotics imitate low-level actions. Demonstrations are typically provided by manually moving the robot [1] or via teleoperation [2]. A critical challenge of this approach is scaling: demonstrations need to be collected in every new environment. On the other hand, imitation of high-level (semantic) skills has the promise of generalization: demonstrations can be collected in one kitchen and applied to any number of kitchens, eliminating the need to re-demonstrate in every new

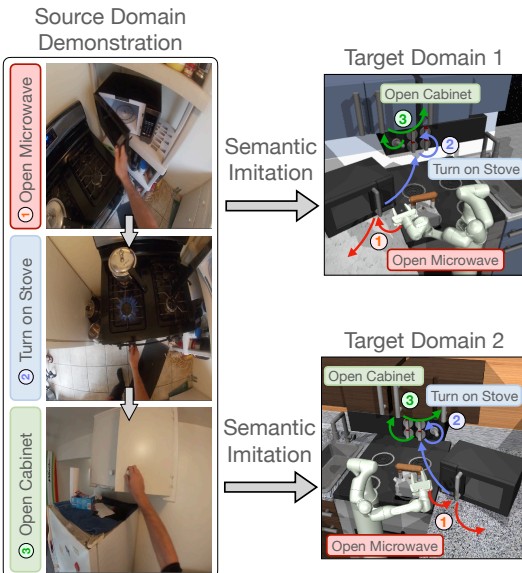

Figure 1: We address *semantic* imitation, which aims to leverage demonstrations from a source domain, e.g., human video demonstrations, to accelerate the learning of the same tasks in a different target domain, e.g., controlling a robotic manipulator in a simulated kitchen environment.

---

[*]Work done during an internship at Meta AI. Correspondence to pertsch@usc.edu.

6th Conference on Robot Learning (CoRL 2022), Auckland, New Zealand.

environment. Learning via imitation of high-level skills can lead to scalable and generalizable robot learning.

In this work, we present **S**emantic **T**ransfer **A**ccelerated **R**L (STAR), which accelerates RL using cross-domain demonstrations by leveraging semantic skills, instead of low-level actions. We consider a setting with significantly different source and target environments. Figure 1 shows an example: a robot arm learns to do a kitchen manipulation task by following a visual human demonstration from a different (real-world) kitchen. An approach that follows the precise arm movements of the human will fail due to embodiment and environment differences. Yet, by following the demonstrated semantic skills like "open the microwave" and "turn on the stove", our approach can leverage demonstrations *despite* the domain differences. Like the chef in the above example, we use prior experience for enabling this semantic transfer. We assume access to datasets of prior experience collected across many tasks, in both the source and target domains. From this data, we learn semantic skills like "open the microwave" or "turn on the stove". Next, we collect demonstrations of the task in the source domain and find "semantically similar" states in the target domain. Using this mapping, we learn a policy to follow the demonstrated semantic skills in semantically similar states in the target domain.

We present results on two semantic imitation problems in simulation and on real-to-sim transfer from human videos. In simulation, we test STAR in: (1) a maze navigation task across mazes of different layouts and (2) a sequence of kitchen tasks between two variations of the FrankaKitchen environment [3]. In both tasks our approach matches the learning efficiency of methods with in-domain demonstrations, despite only using cross-domain demonstrations. Additionally, we show that a human demonstration video recorded within 3 minutes in a real-world kitchen can accelerate the learning of long-horizon manipulation tasks in the FrankaKitchen by hundreds of thousands of robot environment interactions.

In summary, our contributions are twofold: (1) we introduce STAR, an approach for cross-domain transfer via learned semantic skills, (2) we show that STAR can leverage demonstrations across substantially differing domains to accelerate the learning of long-horizon tasks.

## 2 Related Work

**Learning from demonstrations.** Learning from Demonstrations (LfD, Argall et al. [4]) is a popular method for learning robot behaviors using demonstrations of the target task, often collected by human operators. Common approaches include behavioral cloning (BC, Pomerleau [5]) and adversarial imitation approaches [6]. A number of works have proposed approaches for combining these imitation objectives with reinforcement learning [7, 8, 9, 10]. However, all of these approaches require demonstrations in the target domain, limiting their applicability to new domains. In contrast, our approach imitates the demonstrations' *semantic* skills and thus enables transfer *across* domains.

**Skill-based Imitation.** Using temporal abstraction via skills has a long tradition in hierarchical RL [11, 12, 13]. Skills have also been used for the *imitation* of long-horizon tasks. Pertsch et al. [14], Hakhamaneshi et al. [15] learn skills from task-agnostic offline experience [16, 17] and imitate demonstrated skills instead of primitive actions. But, since the learned skills do not capture semantic information, they require demonstrations in the target domain. Xu et al. [18], Huang et al. [19] divide long-horizon tasks into subroutines, but struggle if the two domains requires a different sequence of subroutines, e.g., if skill pre-conditions are not met in the target environment. Our approach is robust to such mismatches without requiring demonstrations in the target domain.

**Cross-Domain Imitation.** Peng et al. [20] assume a pre-specified mapping between source and target domain. [21, 22] leverage offline experience to learn mappings while [23, 24, 25] rely on paired demonstrations. A popular goal is to leverage human videos for robot learning since they are easy to collect at scale. [26, 27] learn reward functions from human demonstrations and Schmeckpeper et al. [28] add human experience to an RL agent's replay buffer, but they only consider short-horizon tasks and rely on environments being similar. Yu et al. [29] meta-learn cross-domain subroutines, but cannot handle different subroutines between source and target. Our approach imitates long-horizon tasks across domains, without a pre-defined mapping and is robust to different semantic subroutines.

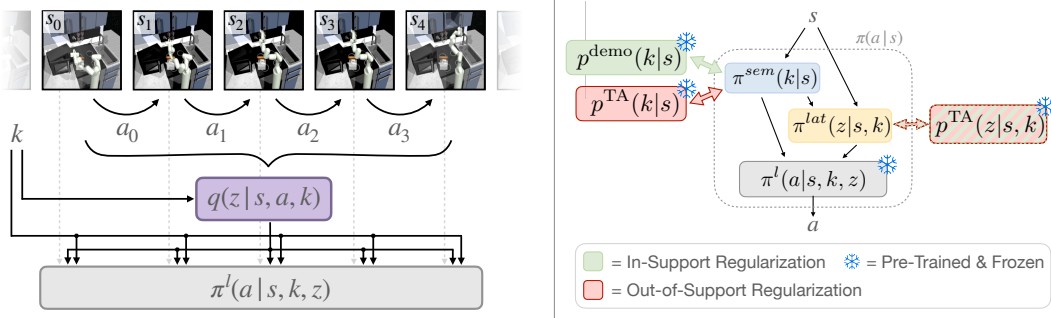

Figure 2: Model overview for pre-training (**left**) and target task learning (**right**). We pre-train a semantic skill policy $\pi^l$ (**grey**) and use it to decode actions from the learned high-level policies $\pi^{sem}$ and $\pi^{lat}$ (**blue** and **yellow**) during target task learning. See training details in the main text.

## 3 Problem Formulation

We define a source environment $S$ and a target environment $T$. In the source environment, we have $N$ demonstrations $\tau_{1:N}^S$ with $\tau_i^S = \{s_0^S, a_0^S, s_1^S, a_1^S, \dots\}$ sequences of states $s^S$ and actions $a^S$. Our goal is to leverage these demonstrations to accelerate training of a policy $\pi(s^T)$ in the target environment, acting on target states $s^T$ and predicting actions $a^T$. $\pi(s^T)$ maximizes the discounted target task reward $J^T = \mathbb{E}_\pi \left[ \sum_{l=0}^{L-1} \gamma^l R(s_l^T, a_l^T) \right]$ for an episode of length $L$. We account for different state-action spaces $(s^S, a^S)$ vs. $(s^T, a^T)$ between source and target, but drop the superscript in the following sections, assuming that the context makes it clear whether we are addressing source or target states. In Section 4.3 we describe how we bridge this environment gap. Without loss of generality we assume that the source and target environments are substantially different; sequences of low-level actions that solve a task in the source environment *do not* lead to high reward in the target environment. In the following we will also use the term *domain* to refer to two environments with this property. Yet, we assume that the demonstrations show a set of *semantic skills*, which when followed in the target environment can lead to task success. Here the term *semantic skill* refers to a high-level notion of skill, like "open the microwave" or "turn on the oven", which is independent of the environment-specific low-level actions required to perform it. We further assume that both source and target environment allow for the execution of the same set of semantic skills.

Semantic imitation requires an agent to understand the semantic skills performed in the demonstrations. We use task-agnostic datasets $\mathcal{D}_S$ and $\mathcal{D}_T$ in the source and target domains to extract such semantic skills. Each $\mathcal{D}_i$ consists of state-action trajectories collected across a diverse range of prior tasks, e.g., from previously trained policies or teleoperation, as is commonly assumed in prior work [16, 17, 14, 15]. We also assume discrete semantic skill annotations $k_t \in \mathcal{K}$, denoting the skill being executed at time step $t$. These can be collected manually, but we demonstrate how to use pre-trained action recognition models as a more scalable alternative (Sec. 5.2).

## 4 Approach

Our approach STAR imitates demonstrations' semantic skills, instead of low-level actions, to enable cross-domain, *semantic* imitation. We use a two-layer hierarchical policy with a high-level that outputs the semantic skill and a low-level that executes the skill.

**Algorithm 1** STAR (Semantic Transfer Accelerated RL)
| |
|---|
| Pre-Train low-level policy $\pi^l(a|s,k,z)$      ▷ cf. Sec. 4.1 |
| Match source demos to target states      ▷ cf. Sec. 4.3 |
| Pre-train $p^{\text{demo}}(k|s), p^{\text{TA}}(k|s), p^{\text{TA}}(z|s,k), D(s)$      ▷ cf. Tab. 1 |
| **for** each target train iteration **do** |
|     Collect online experience $(s, k, z, R, s')$ |
|     Update high-level policies with eq. 3      ▷ cf. Alg. 2 |
| **return** trained high-level policies $\pi^{\text{sem}}(k|s), \pi^{\text{lat}}(z|s,k)$ |

We first describe our semantic skill representation, followed by the low-level and high-level policy learning. Algorithm 1 summarizes our approach.

### 4.1 Semantic Skill Representation

A skill is characterized by both its semantics, i.e., whether to open the microwave or turn on the stove, as well as the details of its low-level execution, e.g., at what angle to approach the microwave or where to grasp its door handle. Thus, we represent skills via a low-level policy $\pi^l(a|s,k,z)$ which is

conditioned on the current environment state $s$, the semantic skill ID $k$ and a latent variable $z$ which captures the execution details. For example, when "turning on the stove", $a$ are the joint velocities, $s$ is the robot and environment state, $k$ is the semantic skill ID of this skill, and $z$ captures the robot hand orientation as it interacts with the stove. A single tuple $(k, z)$ represents a sequence of $H$ steps, since such temporal abstraction facilitates long-horizon imitation [14]. We train our model as a conditional variational autoencoder (VAE) [30] over a sequence of actions *given* a state and semantic skill ID. Thus, the latent variable $z$ represents all information required to reconstruct $a_{0:H-1}$ that is not contained in the skill ID, i.e., information about *how* to execute the semantic skill.

Figure 2, left depicts the training setup for $\pi^l$. We randomly sample an $H$-step state-action subsequence $(s_{0:H}, a_{0:H-1})$ from $\mathcal{D}_T$. An inference network $q(z|s, a, k)$ encodes the sequence into a latent representation $z$ conditioned on the semantic skill ID $k$ at the first time step. $k$ and $z$ are passed to $\pi^l$, which reconstructs the sampled actions. Our training objective is a standard conditional VAE objective that combines a reconstruction and a prior regularization term:

$$\mathcal{L}_{\pi_l} = \mathbb{E}_q \underbrace{\left[ \prod_{t=0}^{H-1} \log \pi^l(a_t|s_t, k, z) \right]}_{\text{reconstruction}} - \underbrace{\beta D_{\text{KL}}\big(q(z|s_{0:H}, a_{0:H-1}, k), p(z)\big)}_{\text{prior regularization}}. \tag{1}$$

Here $D_{\text{KL}}$ denotes the Kullback-Leibler divergence. We use a simple uniform Gaussian prior $p(z)$ and a weighting factor $\beta$ for the regularization objective [31]. The semantic skill ID $k$ is pre-defined, discrete and labelled, while the latent $z$ is learned and continuous. In this way, our formulation captures discrete aspects of manipulation skills (open a microwave vs. turn on a stove) while being able to continuously modulate each semantic skill (e.g., different ways of approaching the microwave).

## 4.2 Semantic Transfer Accelerated RL

After pre-training the low-level policy $\pi^l(a|s, k, z)$, we learn the high-level policy using the source domain demonstrations. Concretely, we train a policy $\pi^h(k, z|s)$ that predicts tuples $(k, z)$ which get executed via $\pi^l$. Note that unlike prior work [14], our high-level policy outputs both, the semantic skill $k$ *and* the low-level execution latent $z$. It is thus able to choose which semantic skill to execute and tailor its execution to the target domain. Cross-domain demonstrations solely guide the *semantic* skill choice, since the low-level execution might vary between source and target domains. Thus, we factorize $\pi^h$ into a semantic sub-policy $\pi^{\text{sem}}(k|s)$ and a latent, non-semantic sub-policy $\pi^{\text{lat}}(z|s, k)$:

$$\pi(a|s) = \underbrace{\pi^l(a|s, k, z)}_{\text{skill policy}} \cdot \underbrace{\pi^{\text{lat}}(z|s, k)\ \pi^{\text{sem}}(k|s)}_{\text{high-level policy } \pi^h(k, z|s)}. \tag{2}$$

Intuitively, this can be thought of as first deciding *what* skill to execute (e.g., open the microwave), followed by *how* to execute it. We pre-train multiple models via supervised learning for training $\pi^h$: (1) two semantic skill priors $p^{\text{demo}}(k|s)$ and $p^{\text{TA}}(k|s)$, trained to infer the semantic skill annotations from demonstrations and task-agnostic dataset $\mathcal{D}_T$ respectively, (2) a task-agnostic prior $p^{\text{TA}}(z|s, k)$ over the latent skill variable $z$, trained to match the output of the inference network on $\mathcal{D}_T$ and (3) a discriminator $D(s)$, trained to classify whether a state is part of the demonstration trajectories. We summarize all pre-trained components and their *supervised* training objectives in Appendix, Table 1.

We provide an overview of our semantic imitation architecture and the used regularization terms in Figure 2, right. We build on the idea of weighted policy regularization with a learned demonstration support estimator from Pertsch et al. [14] (for a brief summary, see appendix B). We regularize the high-level *semantic* policy $\pi^{\text{sem}}$ (**blue**) towards the demonstration skill distribution $p^{\text{demo}}(k|s)$ when $D(s)$ classifies the current state as part of the demonstrations (**green**). For states which $D(s)$ classifies as outside the demonstration support, we regularize $\pi^{\text{sem}}$ towards the task-agnostic prior $p^{\text{TA}}(k|s)$ (**red**). We *always* regularize the non-semantic sub-policy $\pi^{\text{lat}}(z|s, k)$ (**yellow**) towards the task-agnostic prior $p^{\text{TA}}(z|s, k)$, since execution-specific information cannot be transferred across

domains. The overall optimization objective for $\pi^h$ is:

$$\mathbb{E}_{\pi^h}\bigg[\tilde{r}(s,a) \underbrace{-\alpha_q D_{\text{KL}}\big(\pi^{\text{sem}}(k|s), p^{\text{demo}}(k|s)\big)\cdot D(s)}_{\text{demonstration regularization}} \underbrace{-\alpha_p D_{\text{KL}}\big(\pi^{\text{sem}}(k|s), p^{\text{TA}}(k|s)\big)\cdot(1-D(s))}_{\text{task-agnostic semantic prior regularization}},$$

$$\underbrace{-\alpha_l D_{\text{KL}}\big(\pi^{\text{lat}}(z|s,k), p^{\text{TA}}(z|s,k)\big)}_{\text{task-agnostic execution prior regularization}}\bigg]. \tag{3}$$

$\alpha_q$, $\alpha_p$ and $\alpha_l$ are either fixed or automatically tuned via dual gradient descent. We augment the target task reward using the discriminator $D(s)$ to encourage the policy to reach states within the demonstration support: $\tilde{r}(s,a) = (1-\kappa)\cdot R(s,a) + \kappa\cdot\big[\log D(s) - \log\big(1-D(s)\big)\big]$. In the setting with no target environment rewards (pure imitation learning), we rely solely on this discriminator reward for policy training (Section D). For a summary of the full procedure, see Algorithm 2.

The final challenge is that the discriminator $D(s)$ and the prior $p^{\text{demo}}(k|s)$ are trained on states from the source domain, but need to be applied to the target domain. Since the domains differ substantially, we cannot expect the pre-trained networks to generalize. Instead, we need to explicitly bridge the state domain gap, as described next.

### 4.3 Cross-Domain State Matching

Our goal is to find semantically similar states between the source and the target environment. These are states with similar distributions over likely semantic skills. E.g. if the agent's hand is reaching for the handle of a closed microwave, the probability for the skill "open microwave" is high, while the probability for other skills, e.g. "turn on stove" is low. Crucially, this is true independent of the domain and independent of whether e.g. a human or robot is executing the action. Thus, we can use the skill prior distributions to find *semantically* similar states.

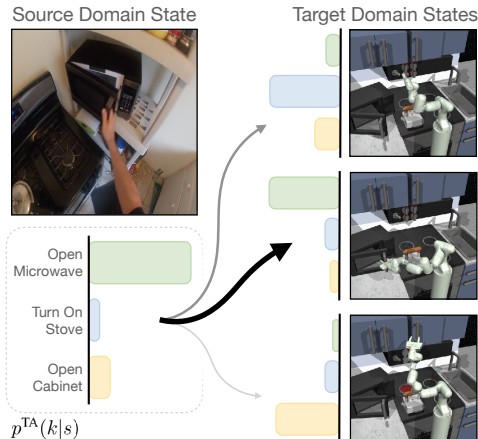

Figure 3: State matching between source and target domain. For every source domain state from the demonstrations, we compute the task-agnostic semantic skill distribution $p^{\text{TA}}(k|s)$ and find the target domain state with the most similar semantic skill distribution from the task-agnostic dataset $\mathcal{D}_T$. We then relabel the demonstrations with these matched states from the target domain.

Following this intuition, we find corresponding states based on the similarity between the task-agnostic semantic skill prior distributions $p^{\text{TA}}(k|s)$. We illustrate an example in Figure 3: for a given source demonstration state $s^S$ with high likelihood of opening the microwave, we find a target domain state $s^T$ that has high likelihood of opening the microwave, by minimizing the symmetric KL divergence between the task-agnostic skill distributions (we omit $(\cdot)^{\text{TA}}$ for brevity):

$$\min_{s^T\in\mathcal{D}_T} D_{\text{KL}}\big(p_T(k|s^T), p_S(k|s^S)\big) + D_{\text{KL}}\big(p_S(k|s^S), p_T(k|s^T)\big) \tag{4}$$

In practice, states can be matched incorrectly when the task agnostic dataset chooses one skill with much higher probability than others. In such states, the divergence in equation 4 is dominated by one skill, and others are ignored, causing matching errors. Using a state's temporal context can result in more robust correspondences by reducing the influence of high likelihood skills in any single state. We compute an aggregated skill distribution $\phi(k|s)$ using a temporal window around the current state:

$$\phi(k|s_t) = \frac{1}{Z(s)}\bigg(\sum_{i=t}^{T}\gamma_+^i p(k|s_i) + \sum_{j=1}^{t-1}\gamma_-^{t-j} p(k|s_{t-j})\bigg) \tag{5}$$

Here, $\gamma_+, \gamma_- \in [0,1]$ determine the forward and backward horizon of the aggregate skill distribution. $Z(s)$ ensures that the aggregate probability distribution sums to one. Instead of $p^{\text{TA}}$ in equation 4, we use $\phi(k|s)$. By matching all source-domain demonstrations states to states in the target domain via $\phi(k|s)$, we create a proxy dataset of target state demonstrations, which we use to pre-train the models $p^{\text{demo}}(k|s)$ and $D(s)$. Once trained, we use them for training the high-level policy via equation 3.

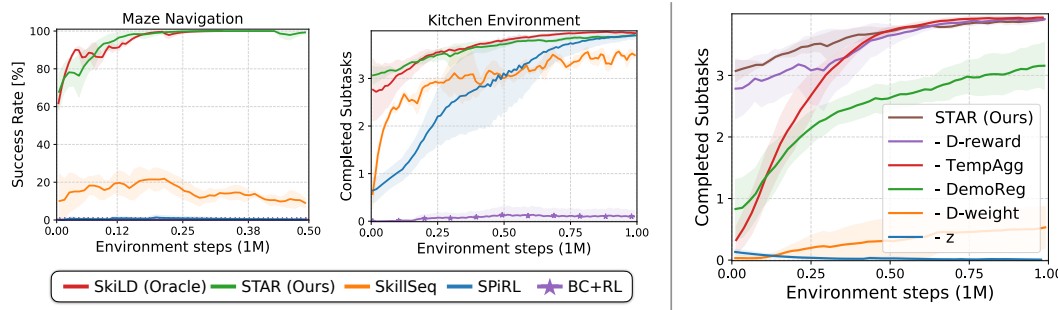

Figure 5: **Left**: Performance on the simulated semantic imitation tasks. STAR, matches the performance of the oracle, SkiLD, which has access to target domain demonstrations and outperforms both SPiRL, which does not use demonstrations, and SkillSeq, which follows the demonstrated semantic skills sequentially. **Right**: Ablations in the kitchen environment, see main text for details.

# 5 Experiments

Our experiments are designed to answer the following questions: (1) Can we leverage demonstrations *across domains* to accelerate learning via semantic imitation? (2) Can we use semantic imitation to teach a robot a new task from real-world videos of humans performing the task? (3) Is our approach robust to missing skills in the demonstrations? We test semantic imitation across two simulated maze and kitchen environments, as well as from real-world videos of humans to a simulated robot. Our results show that our approach can accelerate learning from cross-domain demonstrations, even with real-to-sim gap.

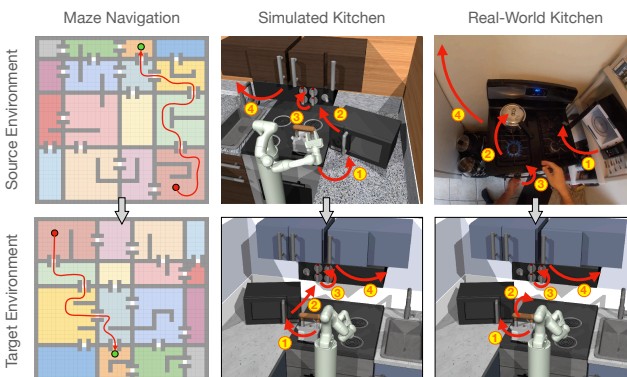

Figure 4: We evaluate on three pairs of source (**top**) and target (**bottom**) environments. **Left**: maze navigation. The agent needs to follow a sequence of colored rooms (red path) but the maze layout changes substantially between source and target domains. **Middle**: kitchen manipulation. A robotic arm executes a sequence of skills, but the layout of the kitchens differs. **Right**: Same as before, but with human demonstrations from a real-world kitchen.

## 5.1 Cross-Domain Imitation in Simulation

We first test our approach STAR in two simulated settings: a maze navigation and a robot kitchen manipulation task (see Figure 4, left & middle). In the **maze navigation task**, both domains have corresponding rooms, indicated by their color in Figure 4. The agent needs to follow a sequence of semantic skills like "go to red room", "go to green room" etc. In the **kitchen manipulation task**, a Franka arm tackles long-horizon manipulation tasks in a simulated kitchen [3]. We define 7 semantic skills, like "open the microwave" or "turn on the stove" in the source and target environments. In both environments we collect demonstrations in the source domain, and task-agnostic datasets in both the source and target domains using motion planners and human teleoperation respectively. For further details on action and observation spaces, rewards and data collection, see Sec C.4.

We compare our approach to multile prior skill-based RL approaches with and without demonstration guidance: **SPiRL** [16] learns skills from $\mathcal{D}_T$ and then trains a high-level policy over skills; **BC+RL** [7, 8] pre-trains with behavioral cloning and finetunes with SAC [32]; **SkillSeq**, similar to Xu et al. [18], sequentially executes the ground truth sequence of semantic skills as demonstrated; **SkiLD** [14] is an oracle with access to demonstrations *in the target domain* and follows them using learned skills. For more details on the implementation of our approach and all comparisons, see appendix, Sections C.1 - C.3.

Figure 5, left, compares the performance of all approaches in both tasks. BC+RL is unable to leverage the cross-domain demonstrations and makes no progress on the task. SPiRL is able to learn the kitchen manipulation task, but requires many more environment interactions to reach the same performance

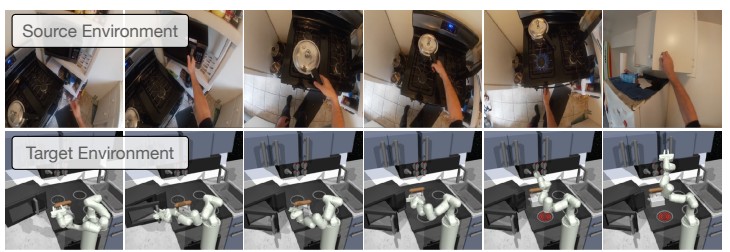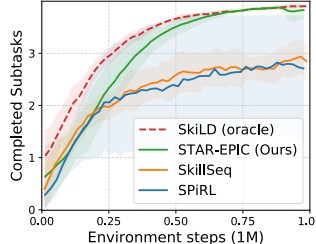

Figure 6: Semantic imitation from human demonstrations. **Left**: Qualitative state matching results. The top row displays frames subsampled from a task demonstration in the human kitchen source domain. The bottom row visualizes the states matched to the source frames via the procedure described in Section 4.3. The matched states represent corresponding *semantic* scenes in which the agent e.g., opens the microwave, turns on the stove or opens the cabinet. **Right**: Quantitative results on the kitchen manipulation task from human video demonstrations.

as our approach. SkillSeq succeeds in approximately 20% of the maze episodes and solves on average 3 out of 4 subtasks in the kitchen manipulation environment after fine-tuning. The mixed success is due to inaccuracies in execution of the skill policies since SkillSeq follows the ground truth sequence of high-level skills. Our approach, STAR, can use cross-domain demonstrations to match the learning efficiency of SkiLD (oracle) that has access to *target domain* demonstrations. This shows that our approach is effective at extracting useful information from cross-domain demonstrations. During downstream task training of the high-level semantic and execution policies our approach can fix both, errors in the high-level skill plan and the low-level skill execution. The ability to jointly adapt high-level and low-level policies and e.g. react to failures in the low-level policy rather than following a fixed high-level plan is crucial for good performance on long-horizon tasks. We find that this trend holds even in the "pure" imitation learning (IL) setting without environment rewards, where we solely rely on the learned discriminator reward to guide learning (see appendix, Section D for detailed results). Thus, STAR can be used both, as a demonstration-guided RL algorithm and for cross-domain imitation learning. Qualitative results can be viewed at https://tinyurl.com/star-rl and in Figure 8.

To study the different components of our approach, we run ablations in the FrankaKitchen environment (Fig. 5, right). Removing the discriminator-based weighting for the demonstration regularization (**-D-weight**) (Eq. 4) or removing the demonstration regularization altogether (**-DemoReg**), leads to poor performance. In contrast, removing the discriminator-based dense reward (**-D-reward**) or temporal aggregation during matching (**-TempAgg**) affects learning speed but has the same asymptotic performance. Finally, a model without the latent variable $z$ (**-z**) cannot model the diversity of skill executions in the data; the resulting skills are too imprecise to learn long-horizon tasks. We show qualitative examples of the effect of varying matching window sizes $[\gamma^-, \gamma^+]$ on the project website: https://tinyurl.com/star-rl.

## 5.2 Imitation from Human Demonstrations

In this section we ask: can our approach be used to leverage human video demonstrations for teaching new tasks to robots? Imitating human demonstrations presents a larger challenge since it requires bridging domain differences that span observation spaces (from images in the real-world to low-dimensional states in simulation), agent morphologies (from a bimanual human to a 7DOF robot arm), and environments (from the real-world to a simulated robotic environment). To investigate this question, we collect 20 human video demonstrations in a real-world kitchen, which demonstrate a task the robotic agent needs to learn in the target simulated domain. Instead of collecting a large, task-agnostic dataset in the human source domain and manually annotating semantic skill labels, we demonstrate a more scalable alternative: we use an action recognition model, pre-trained on the EPIC Kitchens dataset [33], zero-shot to predict semantic skill distributions on the human demonstration videos. We define a mapping from the 97 verb and 300 noun classes in EPIC Kitchens to the skills present in the target domain and then use our approach as described in Section 4.2, using the EPIC skill distributions as the task-agnostic skill prior $p^{\text{TA}}(k|s)$. For data collection details, see Section C.4.

We visualize qualitative matching results between the domains in Figure 6, left. We successfully match frames to the corresponding *semantic* states in the target domain. In Figure 6, right, we show that this leads to successful semantic imitation of the human demonstrations. Our approach

STAR with EPIC Kitchens auto-generated skill distributions is able to reach the same asymptotic performance as the oracle approach that has access to target domain demonstrations, with only slightly reduced learning speed. It also outperforms the SkillSeq and SPiRL baselines (for qualitative results see https://tinyurl.com/star-rl).

To recap: for this experiment we *did not* collect a large, task-agnostic human dataset and we *did not* manually annotate any human videos. Collecting a few human demonstrations in an unseen kitchen was sufficient to substantially accelerate learning of the target task on the robot in simulation. This demonstrates one avenue for scaling robot learning by (1) learning from easy-to-collect human video demonstrations and (2) using pre-trained skill prediction models to bridge the domain gap.

### 5.3 Robustness to Noisy Demonstrations and Labels

In realistic scenarios agents often need to cope with noisy demonstration data, e.g., with partial demonstrations or faulty labels. Thus, we test STAR's ability to handle such noise. First, we test imitation from *partial* demonstrations with missing subskills. These commonly occur when there are large differences between source and target domain, e.g., the demonstration domain might already have a pot on the stove, and starts with "turn on the stove", but in the target domain we need to first place the pot on the stove. We test this in the simulated kitchen tasks by dropping individual subskills from the demonstrations ("w/o Task $i$' in Figure 7). Figure 7 shows that the SkillSeq approach strug-

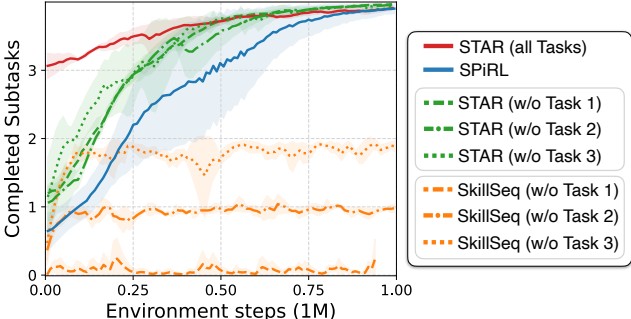

Figure 7: Semantic imitation with missing skills in the demonstrations. Our approach STAR still learns the full task faster than learning without demonstrations (SPiRL), while SkillSeq get stuck at the missing skill.

gles with such noise: it gets stuck whenever the corresponding skill is missing in the demonstration. In contrast, STAR can leverage demonstrations that are lacking complete subskills and still learn faster than the no-demonstration baseline SPiRL. When a skill is missing, the STAR agent finds itself off the demonstration support. Then the objective in equation 3 regularizes the policy towards the task-agnostic skill prior, encouraging the agent to explore until it finds its way (back) to the demonstration support. This allows our method to bridge "holes" in the demonstrations. We also test STAR's robustness to noisy semantic skill labels, in Section E. We find that STAR is robust to errors in the annotated skill lengths and to uncertain skill detections. Only frequent, high-confidence mis-detections of skills can lead to erroneous matches and decreased performance. Both experiments show that STAR's guidance with semantic demonstrations is robust to noise in the training and demonstration data.

## 6 Conclusion and Limitations

In this work, we presented STAR, an approach for imitation based on semantic skills that can use cross-domain demonstrations for accelerating RL. STAR is effective on multiple semantic imitation problems, including using real-world human demonstration videos for learning a robotic kitchen manipulation task. Our results present a promising way to use large-scale human video datasets like EPIC Kitchens [33] for behavior learning in robotics. However, our approach assumes a pre-defined set of semantic skills and semantic skill labels on the training data. We demonstrated how such assumptions can be reduced via the use of pre-trained skill prediction models. Yet, obtaining such semantic information from cheaper-to-collect natural language descriptions of the training trajectories without a pre-defined skill set is an exciting direction for future work. Additionally, strengthening the robustness to skill mis-labelings, e.g., via a more robust state matching mechanism, can further improve performance on noisy, real-world datasets.

**Acknowledgments**

This work was supported by Institute of Information & Communications Technology Planning & Evaluation (IITP) grants (No.2019-0-00075, Artificial Intelligence Graduate School Program, KAIST; No.2022-0-00077, AI Technology Development for Commonsense Extraction, Reasoning, and Inference from Heterogeneous Data) and National Research Foundation of Korea (NRF) grant (NRF-2021H1D3A2A03103683), funded by the Korean government (MSIT).

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
