# OpenReview forum: "Cross-Domain Transfer via Semantic Skill Imitation"
_robot-learning.org/CoRL/2022/Conference — CoRL 2022 Poster_

### Official Review · Reviewer_95b2 · 2022-07-31

**Originality:** Good
**Technical Quality:** Good
**Clarity Of Presentation:** Good
**Impact:** 2

**Recommendation:**

Weak Accept: I recommend accepting the paper, but will not argue for my recommendation if the majority of other reviewers have a different opinion.

**Summary:**

This paper focuses on imitation learning of high-level semantic skills where the source and target domain are different. The paper propose "STAR", a model dealing with cross-domain knowledge of long-horizon skills, based on an assumption that the low-level skills are already learned from the given dataset in both source and target domains. The work outperforms several baseline methods in the aspect of learning speed, result metrics, and no need for demonstrations in the target domain.

**Issues:**

I can capture the main idea and the majority of the model formation of this paper. Due to the lack of knowledge of the related works, however, I cannot easily judge the correctness of the technical details.
But I think the content between line 127 and line 139 can be improved, as I got confused when I read it.

- Give more details to help the understand Eq. 1 and the reasons to support the claims from line 136 to line 139. If the proofs are from related works, please explicitly pointed out where (paper, section, eq, etc.) can I find them.

- Line 129, gives a more proper description of the latent representation z instead of "z captures the robot hand orientation", which may also explain why "A single tuple (k, z) represents a sequence of H steps"

- Why can this inference network help learn the low-level policy network?

Other question:
- What is the difference between the proposed method and a method which is predicting the order of executing skills? E.g. in Figure 4 middle, given some demonstrations, the prediction result is "2, 3, 1, 4", and then the robot just executes the basic skills following the order "skill2"->"skill3"->"skill1"->"skill4", as the low-level skills are already learned.

Typo:
- Table 1 in Appendix, the training obj. should be Eq(1), instead of Eq.(2).

**Quality Of The Limitations Section:**

Limitations are addressed clearly

**Reviewer Expertise:**

3: The reviewer is fairly confident that the evaluation is correct

**Robotics Focus:**

Highly relevant to robotics but no hardware experiments

**Strengths And Weaknesses:**

Pros:
- The motivation and problems to be solved are well explained.
- Good presentation of related works and their limitations.
- The assumptions in the problem formulation are properly made and do not violate common sense.
- Table 1 in the Appendix helps understand different models and their training objectives.
- The experiment results look very interesting and promising, especially the last experiment on learning semantic skills from human demonstrations in a video. The current model solved some of the drawbacks of the related works.
- The limitation section is well written, letting me know that the authors share the same concern as mine, i.e. the existence of pre-defined skills and corresponding data in all the domains.

Cons:
- The approach and model in section 4.1 are not well formulated. From line 113 to line 139, I could not follow the techniques and causality the authors wanted to explain.
- The role of the latent representation z in the current model is not clear. In line 129, the author gives an example saying "z capture the robot hand orientation", but z should have some other meaning, such as a high-level execution description of the task.
- Why a tuple (k,z) can represent a sequence of H steps?
- Model and math lack details and the explanation is not sufficient to help understand the model, e.g., for Eq. 1, I do not understand how did the authors reach this learning objective, and the content from line 135 to line 139 cannot help me understand it, neither the reasons of "capturing the discrete aspects of manipulation skills".
- The model formulation since Section 4.2 becomes easier to understand. However, the entire model seems very complicated. It has a lot of submodules that need rich experience to get trained, which decreases my confidence in reproducing the result by other researchers and allowing the people beyond the authors' group to extend this work.


**Summary Of Recommendation:**

I feel very difficult to determine if this paper should be recommended or not. Thus, I would like to hear other reviewers' opinions. My difficulties come from: (1) I am not familiar with the related works. (2) Most of the presentation of the paper seems correct and the experiments look promising, but the entire model is very complicated. In my opinion, it will be quite hard to be used or extended by other researchers. So I do not feel confident to evaluate the contribution and influence of this paper.

---

> ### Author Response · Authors · 2022-08-24
> **Author Rebuttal**
>
> Thank you for your helpful review! Based on your comments we have improved the description of our skill learning model (Sec. 4.1) and added more detailed descriptions and visualizations of our model’s components and the performed ablation experiments (Sec. F, Fig. 13, Fig. 14).
>
> &nbsp;
>
> **“The model description in Sec. 4.1 is hard to follow. How do you derive the objective in Eq. 1? What does “z” represent?”**
>
> We apologize for the confusion. At an abstract level, our skill learning model is a conditional VAE (Sohn et al, 2015 [1]) that predicts a sequence of actions a_{0:H-1} given a state “s” and a semantic skill ID “k”. Like in a conditional VAE, our objective in Eq. 1 combines an action reconstruction term with a regularization term for the latent variable z. z captures information required for reconstructing the demonstrated low-level actions, but not present in the state and semantic skill. Specifically, z captures variance in demonstrations of the same semantic skill in the same state, e.g. _how_ a certain semantic skill should be executed. For example “the robot hand orientation when turning on a stove” as mentioned in the paper (L129), but it can also contain other features, like _which_ of multiple stove top dials to turn on. Note that since we train our conditional VAE on _sequences_ of actions a_{0:H-1}, a single tuple of (k, z) captures information about a whole action sequence, similar to how the latent variable in a stochastic video prediction model captures information about the whole video (e.g. Saxena et al. [2]). We have updated section 4.1 of the manuscript based on the description above.
>
> &nbsp;
>
> **“The model seems complicated. Is it reproducible?”**
>
> We conduct extensive ablation studies of the different components of our approach (Fig. 5) which justify each part of our model and demonstrate their importance for good performance on the cross-domain imitation task. To clarify the performed ablations we add a new figure that visualizes the used and omitted model components for each ablation (Fig. 14, also in attached PDF) and add more detailed descriptions of each ablation in Sec. F. We also provide a new overview figure for all components of our model (Fig. 13) which highlights that most components are pre-trained fully offline with simple supervised learning objectives (Tab. 1). We found training to be robust to different architecture choices, hyperparameters and environments (we use the same pre-training hyperparameters between all experiments in our paper, except regularization weight \beta in Eq. 1). Thus, we believe that our results are transferable to other environments too. Additionally, we list all used hyperparameters in Sections C.1/C.2 and we will open source the code (already included in supplement) and all data used in this project to facilitate easy extensions by other researchers.
>
> &nbsp;
>
> **“How does the method compare to simply following skills in order?”**
>
> Our paper already provides a comparison that tests exactly this, and our method outperforms it (Figure 5, left). The “SkillSeq” baseline follows the sequence of semantic skills as demonstrated, using a trained set of semantic skill policies (L502). In our experiments we find that this method struggles to learn the downstream task efficiently (Fig. 5, 6), since it suffers from an accumulation of errors of the individual skill policies and slow finetuning. Additionally, we show that this simple skill sequencing strategy is unable to learn from suboptimal demonstrations that e.g. lack a skill, which can often happen in the cross-domain imitation setting (Sec. 5.3). In contrast, our approach is still able to learn even from imperfect demonstrations (Fig. 7).
>
> &nbsp;
>
> **“Typo in Table 1.”**
>
> Thank you for pointing this out – you are absolutely correct and we have fixed the reference!
>
> &nbsp;
>
> We hope that this addressed all concerns raised in your review. Please let us know if you need any other clarifications. Otherwise, please consider raising your score accordingly.
>
> &nbsp;
>
> [1] Sohn, Kihyuk, Honglak Lee, and Xinchen Yan, “Learning Structured Output Representation using Deep Conditional Generative Models”, NeurIPS 2015.
>
> [2] Saxena, Ba, Hafner, “Clockwork Variational Autoencoders”, NeurIPS 2021.

---

### Official Review · Reviewer_8M4B · 2022-07-31

**Originality:** Very Good
**Technical Quality:** Very Good
**Clarity Of Presentation:** Very Good
**Impact:** 3

**Recommendation:**

Weak Accept: I recommend accepting the paper, but will not argue for my recommendation if the majority of other reviewers have a different opinion.

**Summary:**

This paper proposes a method to leverage demonstrations collected in a different instance of an environment, to accelerate the learning process of the same tasks in a new instace (e.g. two different kitchens). They do so by assuming the existence of different datasets, one with demonstrations of the full task at a high-level, in a different and non-accessible environment, and a task agnostic dataset collected on the current environment, where the agent can interact with the objects.
The general method is interesting and effective, although there are many moving parts and I wonder if it couldn't be made simpler.

**Issues:**

As described above.

**Quality Of The Limitations Section:**

Additional details required

**Reviewer Expertise:**

3: The reviewer is fairly confident that the evaluation is correct

**Robotics Focus:**

Highly relevant to robotics but no hardware experiments

**Strengths And Weaknesses:**

Stengths:
- The paper tackles an interesting problem: bootstraping reinforcement learning with demonstrations of a long horizon tasks that come from a different instance of the same type of environment. Results show that the method can indeed accelerate the learning phase, and surpasses the performance of many baselines.
- The experiments section is well designed and shows several ablations that help convincing the reader of the benefits of the method.
- Using a real world video of a person solving those tasks can indeed show the potential of such methods to scale to larger datasets composed of humans solving tasks. This data is plentiful online and could possibly enable exponential growth of the capabilities of the robot.
Weaknesses:
- The method is convoluted and has many moving parts. There are several losses that are based on regularising the policy with a prior learned on the existing datasets. These parts could possibly be swapped for normal supervised learning, making it simpler and possibly without sacrificing performance.
- It is unclear if some baselines fail at a high-level or low-level. In line 244 the authors specify that a baseline mostly fails because the low level policies fail, but the plan is correct. This should be better investigated, as learning low level policies should be easier, and generally not the main point of the paper.
- The task is relatively linear in execution, and therefore, as they also show, simply learning or emulating the sequence of high level action can achieve good performance. As specified before, it is unclear if such benchmarks fail at a high-level (difficult to solve) or low-level (easy to solve). Figure 7 shows that if some skills are missing from the demonstration, these techniques indeed fail. However, I believe it would still be possible to learn a good planner with a larger dataset.

**Summary Of Recommendation:**

I suggest to accept the paper in the current form: the idea is interesting, and the problem they tackle is fundamental for the robot learning community. However, I would suggest the authors to minimise the number of moving parts of their algorithm, that can dramatically hurt reproducibility, and show, possibly with more complex robot environments, the benefits of this method for high-level planning than a BC-like planner

---

> ### Author Response · Authors · 2022-08-24
> **Author Rebuttal (1/2)**
>
> Thank you for your thorough review and insightful comments! We address all of the points raised in the review below! We attach a file to our reply that summarizes the new results & visualizations we added to our paper based on your comments for convenience – they are also reflected in the updated manuscript above!
>
> &nbsp;
>
> **“The method has too many moving parts. Try simple supervised learning instead.”**
>
> Our paper shows that all components of our model are important for good performance and straightforward to train. Concretely, (1) our submission performs detailed ablations of all model components (Fig.5, Sec. 5.1), (2) we ran the ablations you suggested and they support our modeling choices, (3) most of our components are trained independently, offline via supervised learning, making them easy to tune. We will detail these points below. To clarify our model, we also add a new overview figure (Fig. 13) that summarizes all components of our model and their training objectives.
>
> Our submission provides extensive ablation studies for the different components of our method in Fig. 5 (right). To clarify this, we added more detailed descriptions and visualizations of the ablations in Sec. F & Fig.14. These ablations show that each part of the objective is important for the strong performance of our approach. E.g. we report results for ablations that remove the prior regularization and others that remove the discriminator-based weighting of the regularization terms. In both cases performance drops significantly (by 25% and 75+% respectively on the robot manipulation task), showing the importance of our modeling decisions. We also note that most of the components of our approach are trained purely with supervised learning (the skill representations, skill priors and discriminator), and only the downstream task policy is trained with RL (see Fig. 13).
>
> In addition to the ablation experiments present in our original paper, we ran another ablation of our method based on the reviewer’s suggestion: we replace the regularization in the policy update (Eq. 3) with supervised learning using the demonstration skill distribution p^{demo}(k | s), weighted with a tuned factor \alpha, similar to Rajeswaran’18, Nair’18 [1, 2]. This led to a performance reduction of 75% percent on the human-video to robotic kitchen manipulation task compared to our approach (see detailed objective & training curves in attached PDF, Sec. G), underlining the importance of the prior regularization components in our policy objective.
>
> We hope that the new experiment, the clearer visualization and our existing ablation studies clarify the different model components and their necessity for good downstream performance. We believe that this addresses the points in the review, but are happy to run further ablations if you have more suggestions!
>
> &nbsp;
>
> **“Show benefits of your approach over a BC-based planner.”**
>
> Thank you for this suggestion! We add results for a method that uses a BC-based planner over the same set of semantic skills our approach uses in our most challenging human → robot transfer setting. We transfer the demonstrations to the target domain with the same matching procedure used by our method (see Sec. 4.3), but remove any downstream policy training with RL introduced by our method. We find that the simple BC planner can only solve ~1 subtask on average. One possible reason are the noisy semantic skill detections obtained from the EPIC Kitchens model that make simple imitation challenging. In contrast, our approach learns to solve all four kitchen manipulation tasks in sequence, highlighting the importance of the RL optimization introduced in our approach for fixing insufficiencies in the BC-trained planner (see attached PDF, Sec. G, for performance comparison). Note that our SkillSeq baseline is an upper bound for the BCPlanner, since SkillSeq is provided with the ground truth sequence of high-level skills (L508). That being said, we agree with the reviewer that we’d expect to see even larger gains of our method over BCPlanner on environments that require more complex and long-horizon high-level planning, like the recently introduced BEHAVIOR [3] or MineDojo [4] benchmarks. This is an exciting direction for future work!

---

> > ### Author Response · Authors · 2022-08-24
> > **Author Rebuttal (2/2)**
> >
> > **“Provide more analysis of high-level vs low-level failures.”**
> >
> > Thanks for highlighting this important point! The experimental evaluations in our paper allow us to disentangle high-level and low-level failures. Specifically, we provide the “SkillSeq” baseline with the _ground-truth_ high-level skill plan (L508), an advantage over our own approach. Thus the final performance deficiency from the perfect score (-25% on the human → robot manipulation task, Fig. 6) for this baseline is directly due to low-level failures. In contrast, STAR is able to adapt the high-level plan _in reaction_ to failures of the low-level policy, and hence improve the performance over a fixed, _ground truth_, high-level plan.
> >
> > In general, in long-horizon tasks, both high- and low- level failures are intertwined, and it is important to learn both policies together. Our approach can learn to correct both high-level and low-level policy failures: (1) it uses RL, guided by the human demonstrations, to learn a high-level policy that improves the high-level plan and reacts to low-level failures and (2) it improves the low-level skill execution itself by modulating the latent skill execution variable z.
> > We integrated this discussion of high-level vs low-level failures into Sec. 5.1!
> >
> > &nbsp;
> >
> > We hope that this addressed all concerns raised in your review. Please let us know if you need any other clarifications. Otherwise, please consider raising your score accordingly.
> >
> > &nbsp;
> >
> > [1] A. Rajeswaran et al., “Learning complex dexterous manipulation with deep reinforcement learning and demonstrations”, RSS 2018.
> >
> > [2] A. Nair et al., “Overcoming exploration in reinforcement learning with demonstrations”, ICRA 2018.
> >
> > [3] S. Srivastara et al., “BEHAVIOR: Benchmark for Everyday Household Activities in Virtual, Interactive, and Ecological Environments.”, CoRL 2021.
> >
> > [4] L. Fan et al., “MineDojo: Building Open-Ended Embodied Agents with Internet-Scale Knowledge.”, 2022

---

### Official Review · Reviewer_UjLs · 2022-08-02

**Originality:** Very Good
**Technical Quality:** Very Good
**Clarity Of Presentation:** Very Good
**Impact:** 4

**Recommendation:**

Strong Accept: I recommend accepting the paper and will argue for my recommendation even if other reviewers hold a different opinion.

**Summary:**

This paper presents an approach for imitation across cross-domain based on semantic skills.
The approach can utilize demonstrations across substantially differing domains to accelerate the learning of long-horizon tasks.  This approach can be used to leverage human video demonstrations for teaching new tasks to the robot  while being robust to noise in training and demonstration data.
The approach is tested across two simulated maze and kitchen environments as well as from real world videos of human to a simulated robot. The results show that the approach can accelerate learning from cross domain demonstrations.

**Issues:**

Analysis of time complexity  and insights about human video to real world manipulation.

**Quality Of The Limitations Section:**

Limitations are addressed clearly

**Reviewer Expertise:**

4: The reviewer is confident but not absolutely certain that the evaluation is correct

**Robotics Focus:**

Highly relevant to robotics but no hardware experiments

**Strengths And Weaknesses:**

Strengths:
The paper is very well written and organized. The motivation and problem setup is very clear.
The approach presents a promising way to use large scale video datasets for behavior learning in robotics.

Weakness:
The following points could be addressed:
What are the computational challenge of learning cross domain skill? e.g. Time for  episodes?
How would it transfer to real robot manipulation from a human video demonstration?

Update:
The complexity of algorithm is well addressed.


**Summary Of Recommendation:**

The paper has interesting ideas for scalable and generalizable robot learning with relevant experimental results.
I would recommend accept.

---

> ### Author Response · Authors · 2022-08-24
> **Author Rebuttal**
>
> We thank reviewer UjLs for their review and the raised questions. We address all of them below!
>
> &nbsp;
>
> **“What are the computational challenges of your approach?”**
>
> Thank you for this question! The computational complexity of our approach is comparable to that of prior skill-based RL approaches (e.g. SPiRL, Pertsch et al. 2020). Since we have a source and a target domain, we need to train _two_ skill priors, but the size of our models and required computational complexity is still small: we are able to pre-train the skill representation and priors on a single NVIDIA V100 GPU in <10h. Downstream RL training times vary by environment, but are all under 12 hours on the same hardware. We have added a discussion of the computational requirements to sections C1 and C2 of the updated manuscript.
>
> &nbsp;
>
> **“Could we transfer from human video to real robots?”**
>
> For our robotic experiments we use a high-fidelity physics simulator and realistic robot models. While direct transfer of the policies trained in simulation onto a real robot system would be challenging, we believe that our results are indicative of real-robot performance. A major challenge in transferring to real robot systems would be to run downstream RL on real robot hardware, since interactions with the environment would be slow and costly. An interesting alternative could be to replace the online RL algorithm used in our approach with an offline RL method (see e.g. [1, 2]). This could substantially reduce the required amount of environment interactions, while benefiting from semantic demonstrations. This is an exciting direction for future work!
>
> &nbsp;
>
> We hope that this addressed all concerns raised in your review. Please let us know if you need any other clarifications. Otherwise, please consider raising your score accordingly.
>
> &nbsp;
>
> [1] A. Nair, M. Dalal, A. Gupta, and S. Levine. Accelerating online reinforcement learning with offline datasets. 2020.
>
> [2] H. Walke, J. Yang, A. Yu, A. Kumar, J. Orbik, A. Singh, S. Levine. Don't Start From Scratch: Leveraging Prior Data to Automate Robotic Reinforcement Learning. 2022.

---

### Official Review · Reviewer_YQ7o · 2022-08-05

**Originality:** Very Good
**Technical Quality:** Very Good
**Clarity Of Presentation:** Very Good
**Impact:** 4

**Recommendation:**

Strong Accept: I recommend accepting the paper and will argue for my recommendation even if other reviewers hold a different opinion.

**Summary:**

This paper proposed a skill transfer method which exploits learned semantic correspondence in order to transfer skills from source domains to novel target domains. Although the means of executing actions to achieve skills might be very different in terms of low level control, the proposed method can transfer the skill from source domain to the target domain with high performance. The method was verified in two simulation environments and with real human video demonstration data. The method was compared against SOTA methods that do not use semantic information and shown to exhibit superior performance.


**Issues:**

In Eq. 4. I did not understand how aligning p_t(k | s^T) and p_s(k | s^S) helps transfer from source to target since s^T and s^S are not paired. In a sense, we align these two distributions but not necessarily align if they output the same k for semantically same s^T and s^S. I think Eq. 5 partially helps for this issue by also considering multiple timesteps, but I am not sure if this is generic. A clarification helps.

**Quality Of The Limitations Section:**

Limitations are addressed clearly

**Reviewer Expertise:**

3: The reviewer is fairly confident that the evaluation is correct

**Robotics Focus:**

Sufficient demonstration on hardware

**Strengths And Weaknesses:**

- Especially, generalization using real human video demonstrations and automatic extraction of skill labels are very impressive.
- The paper is well-written but a heavy read. A lot of details were only provided in the Appendix, which makes reading the paper challenging. Additionally, dropping the notation for the source/target domain reduces readability.
- The assumption of predefined skill annotations is a very strong assumption. Yet, if the demonstrations/dataset is from robot motions, it is acceptable because these movements are programmed/learned for some purpose, which might be used as the label of the corresponding segment. Additionally, the authors convincingly showed the use of action recognition models in automatic labeling. What is left is the assumption of having "the same set of semantic skills" in between different (source and target) domains. The authors might discuss what happens if the semantic skill set is not exactly the same but similar.
- Related to the previous comment, in order to match the semantic skills, one way might be to focus on the changes the robot creates in its environment. Even though the changes are not exactly the same with different embodiments and arrangements, still the relative changes created on objects by the end effector of the robot would be similar across different domains/environments.
- The paper was shown to have superior performance against baselines that do not use semantic information. I am curious how it would perform if such information is included in other skill transfer methods, for example, in ACNMP which is a demo+RL method that can efficiently transfer skills between different embodiments.


**Summary Of Recommendation:**

The approach is novel, the method is sound, and the methods are verified rigorously in simulation experiments and real data. The paper works on a well-motivated problem. It opens a possibility for a scale-up if we can indeed extract information from demonstrations available in other domains.

---

> ### Author Response · Authors · 2022-08-24
> **Author Rebuttal**
>
> We thank Reviewer YQ7o for their thorough review and helpful comments! We address each of them below!
>
> &nbsp;
>
> **“Baselines do not use semantic skill information. How would a baseline with semantic skills perform?”**
>
> Our “SkillSeq'' baseline *does* use semantic skill information: it follows the sequence of semantic skills from the demonstration using pre-trained semantic skill policies.
> We show in our experiments that although this approach uses semantic skill information, it has substantially lower performance than our approach (Fig. 5) and is more susceptible to incomplete demonstrations (Fig. 7).
>
> &nbsp;
>
> **“How does aligning p_S(k | s^S) and p_T(k | s^T) help transfer between source and target states in Eq. (4)?”**
>
> Intuitively, the semantic skill priors p_S(k | s^S) and p_T(k | s^T) capture the distribution over skills that can likely be executed in a given state and this distribution is similar for semantically similar states. E.g. if the agent’s hand is reaching for the handle of a closed microwave, the prior probability for the skill “open microwave” is high, while the probability for other skills, e.g. “turn on stove” is low. Crucially, this is true independent of the particular domain and independent of whether a human or robot is executing the action. Thus, we can use the skill prior distributions to find _semantically_ similar states, i.e. states in which the agent is likely to open the microwave but not to turn on the stove. By matching the skill prior distributions between source and target domain (Eq. 4) and applying some temporal smoothing (Eq. 5), our approach can robustly match between source and target domain and thereby enable cross-domain imitation. We have expanded our explanation of this in Sec. 4.3 to make this point clearer!
>
> &nbsp;
>
> **“How to handle differing sets of semantic skills in source and target domain?”**
>
> Indeed, we assume matching skill sets between source and target domain in most of our experiments. However, our method is capable of handling differing skill sets. We demonstrate one such scenario in our human → robot experiments with the EPIC kitchens model: we map the 400+ human skills defined in EPIC to the much smaller robot skill set in our target simulated kitchen domain (see Sec. C4). While we use domain expertise to define a mapping, future work can investigate using other approaches, like a pre-trained language model to automatically find a (noisy) mapping between skills by finding nearest neighbors in the learned embedding space.
>
> &nbsp;
>
> **“Can we use state changes to find a mapping between source and target skills?”**
>
> Thanks for this interesting suggestion! Indeed, it should be possible to match skills by observing how they change the semantic state of the environment. A core challenge with this approach would be to obtain a _semantic_ representation of the environment’s state, e.g. whether the microwave is opened vs. closed and the stove turned on vs turned off. Especially in the human video domain where we are only given a pixel representation of the environment state, extracting a semantic state representation might require additional annotations on the training dataset. This is an interesting direction for future work!
>
> &nbsp;
>
> We hope that this addressed all concerns raised in your review. Please let us know if you need any other clarifications. Otherwise, please consider raising your score accordingly.

---

### Author Response · Authors · 2022-08-24
**Updated Manuscript**

Attached is an updated version of our manuscript based on the reviewers’ comments. All changes to the original manuscript are marked in red. Concretely, we:
- Updated Sec. 3, “Problem Formulation”, to clarify our use of the term “domain”
- Expanded our model description in Sec. 4.1
- Provided a better intuition for our state matching procedure in Sec. 4.3
- Added discussion of high-level vs low-level policy failures in the experimental section 5.1
- Expanded our discussion of compute & time complexity in Sec. C1 & C2
- Added a new overview figure of all components of our model (Fig. 13) and detailed descriptions of all performed ablations in Sec. F
- Added results for additional ablations requested by the reviewers in Sec. G

---

### Meta-Review · Area_Chair_o9jz · 2022-08-12

**Recommendation:** Accept (Poster)
**Confidence:** 4

**Metareview:**

The paper is well written and organized, motivation and problem setup clearly stated, tackling an interesting and fundamental problem of bootstrapping RL with information from a different problem instance, and showing promising results, especially when learning from human videos.

However, there are several concerns raised that should be considered and clarified. Mainly regarding the explanation of the complex method, impact of assumptions, and baseline comparisons. See the following proposed action items based on the provided reviews (more details in the corresponding reviews).

Proposal of action items:
- Clarifying the notion of ‘domain’ as a different kitchen is still a kitchen domain, a different domain would be something like a workshop or factory. Also, e.g., L91 source and target ‘environments’ are used, and then switching to ‘domains’. Both wordings appear in several section of the paper and should be clarified.
- Discussing the assumption that semantic skill sets have to be the same for transfer
- Consider a baseline method that also includes semantic information
- Consider an analysis of time and computation complexity
- Clarifying the failures of the baselines (high- vs. low-level)
- Clarifying the approach and modelling descriptions
- Discussing the consequence of the complex model design involving many different parts, pretraining etc. Maybe having additional ablation studies to investigate their influence.

------

Following the consensus recommendations of the reviewers, and enhanced by the additional updates and clarifications of the paper during the rebuttal, my recommendation is to accept the paper.

**Best Paper Nomination:**

No

---

> ### Author Response · Authors · 2022-08-24
> **Author Rebuttal**
>
> Thank you for the review summary. We have addressed each of the reviewers’ points individually and provide pointers for the individual responses below. We added two more ablation experiments that the reviewers asked for and provided new visualizations to clarify our model and ablations. We have also adapted the writing of our paper wherever appropriate to address the reviewers’ concerns.
>
> - **Discussing the assumption that semantic skill sets have to be the same for transfer**
>     - This is not completely correct. Our paper shows experiments where a larger set of human skills (from the EPIC dataset) is used to guide a robot with a smaller set of skills, by mapping the larger set to the smaller one. In the response to Reviewer YQ7o, we clarify this point, and discuss ideas for alternate ways to accommodate a scenario where there might be different skill sets between source and target environment.
>
> - **Consider a baseline method that also includes semantic information**
>     - Our Skill-Seq baseline _uses_ semantic information, by following the sequence of semantic skills demonstrated.
>
> - **Consider an analysis of time and computation complexity**
>     - We added the requested discussion of time / compute complexity in our response to Reviewer UjLs.  A summary of this can also be found in the Appendix, Sec. C1 & C2.
>
> - **Clarifying the failures of the baselines (high- vs. low-level)**
>     - We provided further analysis of high- vs. low-level policy failures in our response to Reviewer 8M4B. A summary can be found in Sec. 5.1.
>
> - **Clarifying the approach and modeling descriptions**
>     - We have added an additional visualization to describe the different parts of our model, including which parts are trained with supervised learning, and which are trained with RL. We also provide a more detailed description of our skill learning model in the response to Reviewer 95b2 and have updated Sec. 4.1 of the manuscript to make the description clearer.
>
> - **Discussing the consequence of the complex model design involving many different parts, pretraining etc. Maybe having additional ablation studies to investigate their influence.**
>     - We already provide several ablation studies in our paper (Fig. 5, Sec. 5.1) that demonstrate the importance of all parts of our model, which we reiterate during the rebuttal. We also add an additional ablation study as suggested by reviewer 8M4B (Sec. G in the updated manuscript) that shows the benefit of our approach over a supervised regularization approach and a BC-based planner.  Finally, we added two new visualizations: Fig. 13 gives an overview of all model components and highlights that most are trained via supervised learning from offline datasets, leading to easy tuning and reproducibility. Fig. 14 visualizes the models used in our ablation experiments and highlights the differences between our approach and ablations.
>
> - **Clarify difference between ‘domain’ and ‘environment’**
>     - Thank you for highlighting this point. We use the term “environment” as it is commonly used in the RL literature: as a scene that the agent can interact with and receive feedback to learn a task. We use the term “domain” when we want to underline stark differences between environments. Specifically, in cross-domain settings, we expect that direct transfer of demonstrations via imitation of low-level actions is impossible, e.g. transferring between kitchens with different layouts. We have added an explanation of this to the problem formulation section (Sec. 3).